# Nest Ecology and Prey Preference of the Mud Dauber Wasp *Sceliphron formosum* (Hymenoptera: Sphecidae)

**DOI:** 10.3390/insects13121136

**Published:** 2022-12-09

**Authors:** David Yuan, Juliey Beckman, Jaime Florez Fernandez, Juanita Rodriguez

**Affiliations:** 1Australian National Insect Collection, CSIRO National Research Collections Australian, Canberra, ACT 2601, Australia; 2Research School of Biology, Australian National University, Canberra, ACT 2601, Australia

**Keywords:** *Sceliphron formosum*, mud dauber wasps, prey preference, Sphecidae, Salticidae, nest ecology, parasitoids

## Abstract

**Simple Summary:**

Mud dauber wasps, *Sceliphron formosum,* are native to Australia and commonly found in urban areas where they build mud nests on human constructions. Mud nests serve as brooding rooms for their larvae, in which paralysed spiders are provided as food. In this study, over 650 mud nests of *S. formosum* were collected, analysed and examined. We first identified the spider taxa that were provided as larval food and then reared the nests that contained larvae, prepupal larvae and pupae of *S. formosum* and unknown insects. By identifying the insects that emerged as adults, we were able to establish the interspecific interactions and reveal that not only do *S. formosum* larvae use the nests, but parasitoids and any opportunistic insects also utilize the empty nests as shelters. Overall, we documented the prey preference of *S. formosum*, as well as 16 families and 23 species of insects from Hymenoptera, Diptera and Coleoptera that are associated with this mud nest life, suggesting a complicated role these mud nests play in an ecosystem.

**Abstract:**

(1) Background: *Sceliphron* is a genus of sphecid wasps that build mud nests for reproduction. These wasps prey exclusively on spiders, and commonly inhabit human constructions. The nesting behaviour and prey selection of many *Sceliphron* species are well studied, but despite being a common insect in urban areas, *Sceliphron formosum* has never been comprehensively studied. (2) Methods: In this study, over 650 mud nests of *S. formosum* were collected, analysed and examined to establish prey preference, nest ecology and interspecific interactions. Prey preference was evaluated in terms of abundance, diversity and morphology. Preference in terms of morphology was estimated using body length to leg span ratio (BLR). (3) Results: *S. formosum* largely preys on ground-hunting spiders, among which Salticidae represented the most collected prey. In terms of prey size, *S. formosum* captures prey with a large BLR. Moreover, an unexpected discovery showed that the enclosed mud nests provide a micro niche that supports a wide variety of insects. Sixteen families and 23 species of insects were found associated with the use of mud nests, comprising the insect orders Hymenoptera, Diptera and Coleoptera. These included important pollinators, new species and native species not recorded in the past 20 years of mud dauber wasp research. We propose the potential of *S. formosum* as a keystone species, due to its ability to provide a micro niche for native species in urban areas. We also discuss how these results contribute to our knowledge on the role of insects in urban ecosystems and their significance in relation to conservation, ecology and biodiversity studies.

## 1. Introduction

Mud dauber wasps belong to the sphecid genus *Sceliphron* Klug, which contains 35 species, occupying all major biogeographical regions of the world [1]. Three species have been recorded in Australia: the two endemic species *Sceliphron laetum* Smith and *Sceliphron formosum* Smith, and the introduced *Sceliphron caementarium* Drury, from North America [2]. In natural situations, *Sceliphron* nests are constructed on shaded and sheltered substrates, such as rock overhangs, sheltered sites on trees or in hollow logs [2]. While in urban areas, *Sceliphron* wasps are seen commonly inhabiting human constructions, building mud nests under eaves and roofs, along the periphery of windows and in other sites that provide the necessary shelter [3]. The mud nest is the brooding room for their larvae, consisting of one or multiple cells that are provisioned with paralysed spiders [4]. The majority of mud dauber wasps practice mass provisioning, whereby female wasps prepare food for larvae prior to laying their eggs [5].

While all *Sceliphron* species practice similar larval provisioning, their nests may differ in shape and in number of mud cells. Wasps in the subgenus *Sceliphron* construct multiple mud cells that amalgamate into one large mud nest, whereas those in the subgenus *Prosceliphron,* build a single-celled mud nest [6]. Of the three *Sceliphron* species that occur in Australia (*Sceliphron formosum, Sceliphron laetum* and *Sceliphron caementarium*), *S. formosum* is the only member of the subgenus *Prosceliphron* and is the least studied. While *S. laetum* is widespread in Australia, *S. formosum* is more confined to northern and eastern Australia and less spotted elsewhere [7]. Despite their rarity elsewhere, mud nests of *S. formosum* are common in urban areas of the Australian Capital Territory (ACT).

To date, most studies regarding *Sceliphron* wasps are of its prey preference that associates with larval provisioning and wasp–spider interactions. Despite an early study proposing that *Sceliphron* species hunt spiders without any prey preference [8], there is now enough evidence to argue that there is prey preference shown by *Sceliphron*, and it is affected by spider sizes [5], spider taxa [5,9,10,11,12,13,14,15,16], spider defence responses [17] and individual specialization [18]. Among these, the prey preference of spider taxa was the most frequently discussed. Though studies have consistently shown that *Sceliphron* species prefer to prey on orb-web spiders (Araneidae) [5,9,10,11,12,13,14,15,16], a single observation [7] recorded *Sceliphron formosum* having preferred prey largely composed of jumping spiders (Salticidae). To further explore this observation, the first part of our study aimed to record the prey preference of *S. formosum* in ACT, as well as investigating the drivers behind its specific prey preference. 

On the other hand, *Sceliphron* nest ecology has not been studied as much as its prey preference. Only a few studies and observations in the past have recorded that mud nests, despite their breeding purpose, are often exploited by parasitic insects or opportunistic insects that use mud nests for reproduction; most of which are of the *Sceliphron* species under subgenus *Sceliphron,* which build multi-cell mud nests. For example, nests of *S. laetum* have been reported as being parasitised by flies of the families of Bombyliidae and Sarcophagidae, and wasps of the families Eulophidae and Chrysididae; while abandoned nests can be used by wasps of the families Crabronidae and Vespidae [2]. Nests of *S. caementarium* were reported in Italy with their parasitoids, inquilines and parasitoids of the inquilines [19]. In Crimea, a recent study reported that nests of *Sceliphron destillatorium* constitute an important resource for insect species that nest in pre-existing cavities [12]. 

Of great interest are data on the influence of mud nests of *S. formosum* on the native insect fauna. Here, we refer to those insects that use cells of *S. formosum* as “tenants” due to their nature of using the pre-existing mud nests. The second part of our study presents the discovery through the analysis of more than 650 mud nests of *S. formosum* and their contents, uncovering the nest ecology, community dynamics and tenant succession associated with the *S. formosum* mud nest system. 

## 2. Materials and Methods 

### 2.1. Mud Nest Collecting

Potential nest collecting sites (mostly underpasses and bicycle tunnels in suburbs of Canberra, ACT, Australia) were located by the satellite feature of Google Maps. Two *Sceliphron* species, *S. formosum* and *S. laetum*, with distinctly different nest structures, were encountered, but we confined our study to *S. formosum.*

Mud nests were collected from August to October 2018 at the selected underpasses and bicycle tunnels throughout the ACT region in Australia (Figure 1). Every site was approximately 2–4 m in height, and the majority of nests could be reached with a standard 1.8 m step ladder. Mud nests were found either sealed by *S. formosum* or other insect tenants and containing live specimens, open and abandoned, or in the process of being provisioned by nest users, containing prey items or completely empty. An angled pallet knife was used to remove mud nests from the attached substrate. Every reachable nest was taken down and sealed nests were brought to the lab for further analysis. A total of 698 nests of *S. formosum* were collected, of which 655 contained live insects and 43 contained dead spider prey.

### 2.2. Examination of Nest Contents 

Unknown larvae inside mud nests were reared to adults by placing them inside an empty Petri dish at room temperature until emergence, when they were euthanised at −20 °C and mounted using insect pins. Forty-three nests contained dead, nearly untouched spider prey and were analysed for prey preference. Spider body length was measured from the anterior of the cephalothorax to posterior abdomen, and leg span was measured with the longest pair of legs spread out, perpendicularly to the body. After measurement, spiders were preserved in 85% ethanol and identified with the online key to Australian jumping spiders (https://apps.lucidcentral.org/salticidae/text/intro/index.html (accessed on 28 May 2019)).

Specimen images were taken with a Leica DFC500 camera mounted on a Leica M205C microscope. Raw images were then aligned and stacked using the Leica Application Suite (LAS) V4.9. and Helicon Focus 5.3. software. Insect specimens were identified at the family level using the CSIRO online guide to Australian insect families and narrowed down to genera and species level using taxonomic literature [6,20,21,22,23,24,25] and the assistance of entomologists of specialized fields.

### 2.3. Statistical Analyses of Tenant Communities

Coleman rarefaction was used as the richness estimator for the tenant community. Briefly, Coleman represents the number of species expected in the total of samples, assuming individuals are distributed at random among samples [26,27]. The EstimateS software program [28] was used to transform raw data of insect species collection at each site (each sample) into Coleman rarefaction by randomly selecting samples and calculating the average of obtained species numbers, repeatedly until covering all samples. A species accumulation curve was then charted using Microsoft Excel. 

## 3. Results

### 3.1. Survey and Observation of the Nests of Sceliphron formosum in ACT

*Sceliphron formosum* nests are single-celled and shaped like sweet potatoes or yams. A completed nest is approximately 3–3.5 cm long and 1–1.5 cm wide and contains a single cell. All the nests were built in the shade, either sheltered by the entire tunnel, bridge or roof. No nests were exposed to direct sunlight. Most nests were built on smooth cement or wood substrates, and located on the wall of residential houses, under bridges, underpasses or tunnels. Although nests can be found on almost all human constructions, we selected bicycle tunnels and underpasses, as other sites can be difficult to access or are located at an unreachable height. Moreover, we observed that the density and number of nests are the greatest in underpasses and bicycle tunnels. 

### 3.2. Spider Prey Composition

A total of 43 sealed nests containing uneaten spiders were examined in the laboratory. Each cell contained 4–30 spiders (average = 14) from seven spider families (Table 1). Apart from Araneidae, most other spider prey are ground-hunting spiders with Salticidae being the most commonly collected. While 77% of nests were largely composed of Salticidae spiders, some wasps specialised in other prey families. For example, Araneidae comprised 47% and 54% of prey in two nests, Sparassidae was found at 32% and 75% in two nests and Hersiliidae at 62% and 60% in two nests. 

Among the most abundant spider prey, 12 genera of Salticidae were found (Table 2) with the majority (43.4%) of the spiders from the genus *Opisthoncus*, and 30.6% from the genus *Servaea.* Less abundant genera were 14.3% *Cytaea*, 3% *Helpis*, 2.6% *Simaethula*, 1.5% *Holoplatys*, 1.1% *Simaetha*, 0.4% *Zenodorus*, 0.4% *Bianor*, 0.4% *Sandalodes*, 0.4% *Clynotis* and 1.9% unknown genera. 

### 3.3. Body Length to Leg Span Ratio (BLR) 

Spider size (tip of head to tip of abdomen) ranged from 0.15 to 1.10 cm, (mean = 0.49 cm, SD = 0.13). Salticid spiders were the largest in size (mean = 0.49 cm, SD = 0.09), while other families ranged from 0.28 to 0.35 cm (mean = 0.30 cm, SD = 0.14) (Figure 2).

Here, we use a novel estimator, BLR, which is calculated as body length/leg span. Among spiders of the same body length, the larger BLR ratio represents a shorter leg span. Salticidae prey had BLR ranging from 0.31 to 0.5 (mean = 0.42, SD = 0.055), which was the largest among all families, followed by Araneidae with BLR ranging from 1.15 to 1.95 (mean = 0.26, SD = 0.023), Sparassidae with BLR ranging from 0.18 to 0.26 (mean = 0.21, SD = 0.025) and Hersiliidae with BLR ranging from 0.09 to 0.2 (mean = 0.16, SD = 0.026) (Figure 3).

### 3.4. Nest Composition 

Over three months, 655 nests that contained live insects at 53 nesting sites were sampled and 23 species of insects were found to be associated with the use of mud nests. Within the sampling range, the species accumulation curve almost reached the asymptote (Figure 4), implying that the number of nests collected may explain all possible insect diversity. This suggests that within the sampling period, we have closely recorded the greatest richness of insects that are associated with the use of the *Sceliphron* mud nest.

Sixteen insect families were divided into seven groups according to their nest use (Figure 5; Table 3). A typical life cycle for a mud nest consists of construction by *S. formosum* (1st group) and eclosion followed by occupation by secondary tenants (2nd group). During larval development, unexpected entry can be made by parasitic insects (3rd and 4th groups). Some insects can arrive as by-catch (5th and 6th group). Lastly, scavengers (7th group) can exploit the mud nests that contain prey debris or dead *S. formosum* or tenants. 

In terms of abundance, 24.7% of the specimens were *Sceliphron formosum* nest builders, 58.6% were secondary tenants, 4.9% were parasitoids of the nest builder, 7.5% were parasitoids of secondary tenants, 0.2% were invaders that shared the nest, 2.1% were by-catch tenants and the remaining 5% were scavengers. Based on how these insect species are involved with the life cycle of mud nests, we established a tenant succession map (Figure 6).

The map explains the community dynamics associated with the use of the mud nest, from the time the nest is built (yellow box) to the point where the nest is vacated and occupied by other insect species (grey box), and accompanied by unexpected invasive events (white box) or unexpected death of larva (skull). 

#### 3.4.1. Group 1: Original Nest Builder

##### *Sceliphron formosum* (Smith, 1856) (Hymenoptera: Sphecidae)

This is the only Australian *Sceliphron* species that builds separated, single-celled mud nests without an extra layer of mud cover (Figure 7a). The cocoons of the genus *Sceliphron* are distinctively baseball bat-shaped with a transparent, dark brown colour (Figure 7b), and the pre-pupal larva or pupa is visible inside, which is also distinguishable by the body colour and characteristics of head capsules. Larval and cocoon characters allowed us to distinguish *S. formosum* from other tenants occupying the mud nests (Figure 7b,c).

The lab-reared larvae had an average of 32 days of pupal duration. Adult *S. formosum* are medium-sized wasps with black and yellow maculation (Figure 7d,e) that differ within the genus *Sceliphron*. From the lab-reared specimens of *S. formosum*, two variations were observed. Besides the description from [6] showing that the yellow spots at the apex of the dorsal enclosure may be present or absent, we also observed that yellow maculation on the femur of hind legs may be intercepted with black colour on some individuals.

#### 3.4.2. Group 2: Secondary Tenants That Exploit Abandoned Nests

##### *Pison* spp. (Hymenoptera: Crabronidae)

A large number of mud nests contained insects with unusual cocoon types and in two or more cells in a single nest. These cocoons were similar in shape to those of *S. formosum* and some were transparent. Other cocoons had hardened textures and were opaque. These cocoons were categorized into groups by their cocoon characters and nesting structure inside the original mud nests. Adults that continuously emerged throughout November and December were identified as multiple *Pison* species (Hymenoptera: Crabronidae). The genus *Pison* is known for nesting in pre-existing cavities. Similar to *Sceliphron* spp., *Pison* wasps practice mass provisioning and provide spiders as larval provender. Six *Pison* species have been identified that nest in empty nests of *S. formosum* [23,25]. 

(1)*Pison simillimum* was the dominant tenant found in *S. formosum* nests. Females occupy the nest by building their own cells, which can make up one or two cells in a single nest (Figure 8a,c). The larvae are grub-like, the head capsule is clear, and the body light is yellow with protrusions on the side of segments (Figure 9a). The cocoon is oval, light-brown coloured, with a paper-like texture and a darkened cap that serves as a moulting exit (Figure 8b). The adults are entirely black, but have ferruginous tibia, tarsus and distinctive brown pubescence on the lower rim of each tergite (Figure 10a).(2)*Pison spinolae* is similar to *P. simillimum* but can be distinguished by the pupal case. The cocoon of *P. spinolae* is oval, dark-brown coloured with a crispy eggshell-like texture (Figure 8e). The adult is entirely black with scarce white pubescence (Figure 9b).

**Figure 8 insects-13-01136-f008:**
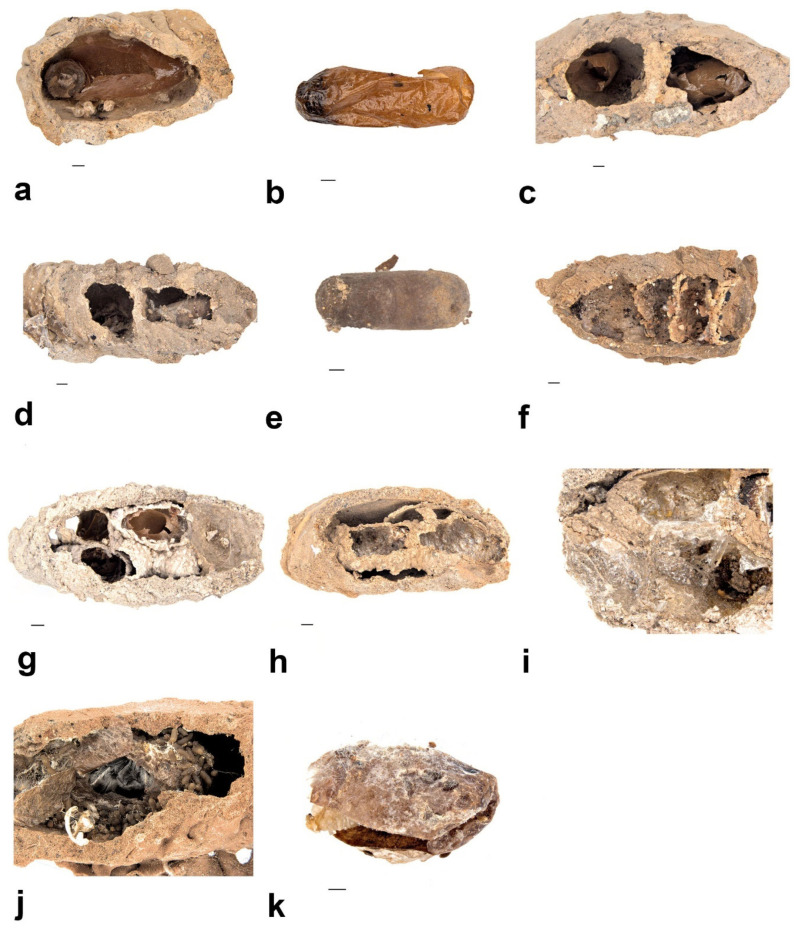
Nest structures and cocoon types of secondary tenants that nest inside mud nests of *S. formosum*: (**a**) *Pison simillimum* mud nest that contains the single cell and cocoon debris; (**b**) empty cocoon of *Pison simillimum;* (**c**) *Pison simillimum* mud nest that contains double cells and cocoon debris (**d**,**e**); nest of *Pison* spp. that produce hard cocoon; (**d**) *Pison auriventre* mud nest; (**e**) cocoon of *Pison spinolae;* (**f**) compartmentalized nests of *Pison priscum* with multiple cells lined up inside; (**g**) compartmentalized nests of *Pison peletieri* with ball-like nests built inside; (**h**) pipe-like nest of *Paralastor* sp. built inside; (**i**) nest of *Hylaes nubilosus,* with visible cellophane coatings that represents the genus *Hylaeus;* (**j**) nest of Megachile, with nearly emerged adult inside of the faeces-covered cocoon; (**k**) Brown cocoon and a dead larva of *M. aurifrons*. Scale bar = 1 mm.

**Figure 9 insects-13-01136-f009:**
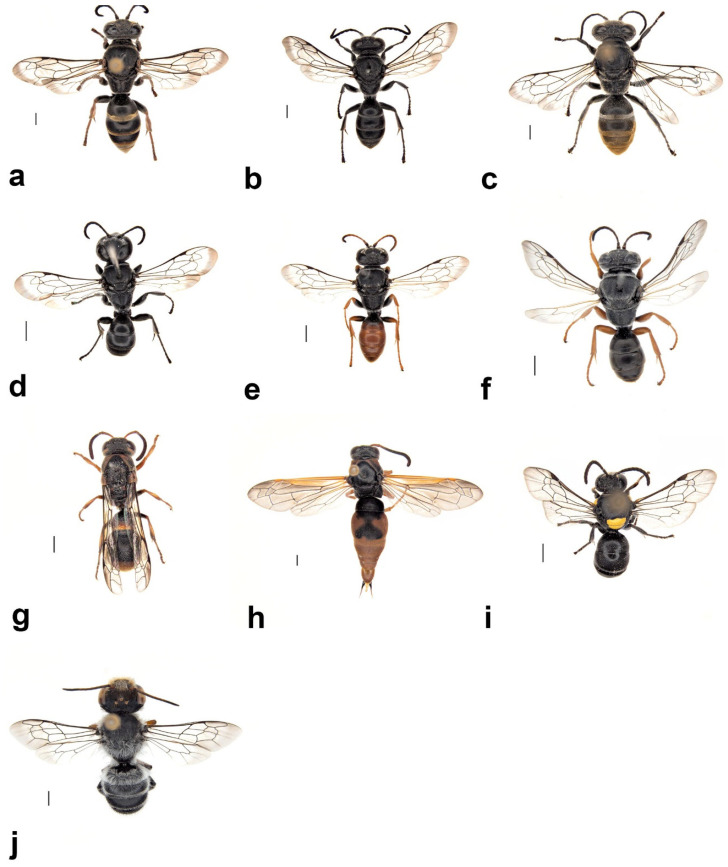
Adults of secondary tenants that are associated with the use of *S. formosum* mud nests: (**a**) *Pison simillimum;* (**b**) *Pison spinolae;* (**c**) *Pison auriventre;* (**d**) *Pison priscum;* (**e**) *Pison peletieri;* (**f**) *Pison prostratum;* (**g**) *Paralastor* sp.; (**h**) unknown species of Eumeninae wasp; (**i**) *Hylaeus nubilosus;* (**j**) *Megachile aurifrons.* Scale bar = 1 mm.

(3)*Pison auriventre* also produces a hard cocoon, but nests were found having two cells (Figure 8d), while *P. spinolae* built one-celled nests within the *S. formosum* mud nest. The adult is entirely black but with golden pubescence on the last three terga (Figure 9c).(4)*Pison priscum* is a small-sized *Pison.* Adults make their own cells inside the nests of *S. formosum.* They usually build two to three cells in one nest, compartmentalized by mud walls (Figure 8f). Cocoon and larvae are similar to *P. simillimum* but smaller in size. Adult is entirely black in colour (Figure 9d).(5)*Pison peletieri* is another small-sized *Pison.* Adults were found building small ball-like mud nests inside empty mud nests (Figure 8g). The cocoon and larvae are similar to *P. simillimum* but smaller in size. The head, thorax and part of the femur of the adult are black, while the rest of the legs are ferruginous (Figure 9e).(6)*Pison prostratum* also build small ball-like nests inside empty nests and have a cocoon similar to that of *P. simillimum*. The main difference between *Pison peletieri* and *Pison prostratum* is body colouration. The adult is entirely black, except for the legs and ventral segments of the antennae (Figure 9f).

##### *Eumeninae wasps* (Hymenoptera: Vespidae)

Two species of potter wasps (Eumeninae) were reared from nests of *S. formosum* in October and November. Both have similar larvae with light yellow body colour and bent body segments, but one constructs a pipe-like nest inside nests of *S. formosum* while the other simply uses the empty nest. Neither produces cocoons. 

*Paralastor* sp. is a small-sized potter wasp (Figure 9g) that builds distinct pipe-like nests inside vacated nests of *S. formosum* (Figure 8h). Usually, two larvae grow inside one pipe, compartmentalized by a silk-like cell wall. Larvae do not produce a pupal case. Pre-pupal larvae have a yellow and bent body.

Eumeninae sp. is a middle-sized potter wasp (Figure 9h) that does not construct its own nests inside the nests of *S. formosum.* Either one or two larvae were found in each single nest. Two larvae are compartmentalized by a silk-like cell wall that is similar to that of the *Paralastor* species.

##### *Hylaeus nubilosus* (Smith, 1853) (Hymenoptera: Colletidae)

Fifteen nests were found containing multiple larvae with cells separated by cellophane wall coatings. Adults emerged during October–December and were found nesting in abandoned nests of *S. formosum* (Figure 9i). Usually, two larvae are found in one cell, compartmentalized by cellophane wall coatings. The grub larvae are pale with white body colour, straight body segments and do not have a head capsule, but have visible mandibles buried at the head (Figure 10b). Larvae produce cellophane coatings covering the mud nests inside and do not produce cocoons (Figure 8i). Adult males and females can be distinguished by the colour pattern at the frons.

**Figure 10 insects-13-01136-f010:**
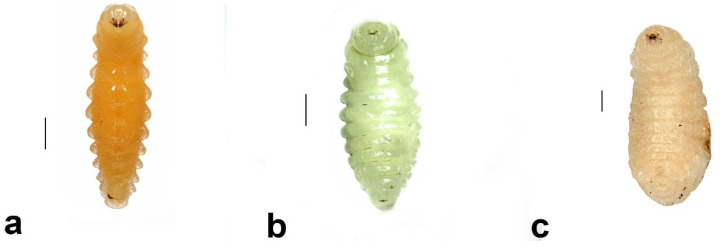
Larvae of secondary tenants: (**a**) larva of *Pison simillimum,* (**b**) larva of *Hylaeus nubilosus,* and (**c**) larva of *Megachile aurifrons.* Scale bar = 1 mm.

##### *Megachile aurifrons* Smith, 1853 (Hymenoptera: Megachilidae)

Two nests contained larvae of Megachilidae from a single site and one male was successfully reared (Figure 9j). The grub larvae are hairy, with pale, white body colour, relatively wider and larger than the larva of *S. formosum* (Figure 10c) and produce a silky brown cocoon covered with its faeces (Figure 8j,k). 

#### 3.4.3. Group 3: Parasitoids of *S. formosum*

##### *Melittobia australica* Girault, 1912 (Hymenoptera: Eulophidae)

Multiple cocoons of *S. formosum*, *Pison* species and *Amobia burnsi* were found dead and contained unknown debris. After examining the debris under the microscope, pupae and adults of *M*. *australica* were found inside. It is clear that *M*. *australica* were parasitising not only *S. formosum*, but also every tenant that produces cocoons during pupal development, including *Pison* spp. and *Amobia burnsi*. Both sexes of *M*. *australica* and debris of host larvae were found inside the cocoons that were parasitised (Figure 11a,b).

##### *Amobia burnsi* (Malloch, 1930) (Diptera: Sarcophagidae)

*Amobia burnsi* were found inside nests with uneaten spider debris (Figure 11d). Larvae of *Amobia burnsi* consume wasp larvae and pupate while surrounded by uneaten spiders. Larvae produce a slim, oval puparium (Figure 11c). All the specimens were reared in the lab; hence, we were unable to observe how *Amobia burnsi* adults break through the mud nests.

#### 3.4.4. Group 4: Parasitoids of Secondary Tenants

##### *Brachymeria* sp. (Hymenoptera: Chalcididae)

Two *Brachymeria* chalcid wasps (Figure 12) were reared from one *Pison* sp. ball-like nest. Larva did not produce a cocoon and occupied one cell from a nest that contains four cells. Interestingly, the same nest is utilized by four different insects: a cuckoo wasp (Hymenoptera: Chrysididae) was found in the first cell, *Gasteruption* sp. (Hymenoptera: Gasteruptiidae) parasitised the next two cells, while *Hylaeus* sp. (Hymenoptera: Colletidae) occupied the last.

##### *Toryminae* sp. (Hymenoptera: Torymidae)

Toryminae chalcid wasps were reared from nests that were occupied by *Paralastor* (Hymenoptera: Vespidae). Six larvae were found from the pipe nest *Paralastor* constructed and four emerged successfully (Figure 13e). The larvae were approximately 0.2 cm, grub-like, with hairy dorsum and no visible head capsule, and did not produce a cocoon (Figure 13c).

##### *Phrudus* sp. (Hymenoptera: Ichneumonidae)

Two *Phrudus* ichneumonid wasps were found in a two-cell nest occupied by *Pison* sp. (Figure 13c). Adults died inside the slender brown pupal case (Figure 12f).

##### *Gasteruption cinerescens* Pasteels, 1957 (Hymenoptera: Gasteruptiidae)

*Gasteruption cinerescens* were reared from ball-like *Pison* sp. nests and *Hylaeus nubilosus* nests (Figure 12d,e). The adult female has a long, slender ovipositor and was reared from a one-celled mud nest, while other males were reared from two to multiple-celled nests (Figure 13d). Larvae are slim in shape and hairy dorsally. 

##### *Primeuchroeus reversus* (Smith, 1874) (Hymenoptera: Chrysididae)

*P. reversus* (Figure 13a) was reared from a *Pison priscum* nest. The larvae produce a red-bean-like cocoon, which occupies the first cell of a *Pison priscum* nest (Figure 12a).

##### *Primeuchroeus faustus* (Smith, 1874) (Hymenoptera: Chrysididae)

*P. faustus* (Figure 13b) was reared from a ball-like *Pison* sp. nest. The larvae produce a cocoon similar to that of *P. reversus* (Figure 12b).

##### *Thraxan* sp. (Diptera: Bombyliidae)

An undescribed species of *Thraxan* beefly was observed parasitising larvae of *Pison simillimum* [29]. 

#### 3.4.5. Group 5: Share Space with *S. formosum*

##### *Epipompilus mirabundus* Yuan & Rodriguez, 2020 (Hymenoptera: Pompilidae)

A white, silky cocoon was discovered within a single mud nest, accompanied by a cocoon of *Sceliphron formosum* with a pre-pupal larva inside. (After a month the white cocoon emerged and was identified as an undescribed *Epipompilus* spider wasp [30]. The *S. formosum* adult emerged shortly afterwards.) Debris of three spider individuals was found, but the debris evidence cannot establish the host association of *Epipompilus.*

#### 3.4.6. Group 6: The by-Catch Tenant

##### *Ogcodes pygmaeus* White, 1914 (Diptera: Acroceridae)

Dead *Oncodes pygmaeus* adults and pupae were found mixed with uneaten spiders inside the nests of *Sceliphron formosum*. We have reported the association of *O. pygmaeus* with two species of jumping spiders [31]. 

#### 3.4.7. Group 7: Scavenger

##### *Anthrenus* sp. (Coleoptera: Dermestidae)

Larvae or adult *Anthrenus* were found feeding in the nests either full of uneaten spider debris or dead pupa of *S. formosum* (Figure 13g). One nest usually contained one larva and all the shed skin from different instars was left in the nest. 

## 4. Discussion

### 4.1. Prey Preference

Previous studies have shown that other *Sceliphron* species prefer to prey on orb weavers, particularly, Araneidae spiders [12,14,15], while a single observation made by Callan [7] reported *Sceliphron formosum* preferentially hunted for Salticidae. Our study further established that *S. formosum* prefers Salticidae spiders over other families, which comprise more than 80% of the spiders found in nests. Moreover, Salticidae and other spider prey found in the nests were ground-hunting spiders, whereas only 6.7% of spider prey were orb weavers, showing that *S. formosum* prefers to collect ground-hunting spiders. This might suggest that the predating behaviour of *S. formosum* might differ from other *Sceliphron* species, as it requires different hunting techniques to capture web weavers or to capture ground, surface foraging spiders. 

Having different prey preferences among sympatric *Sceliphron* species in the ACT may be beneficial because of the reduced competition. An example of congener competitive exclusion reported by Fateryga and Kovblyuk [12] showed that the abundance of *S. destillatorium* noticeably decreased due to the immigration of *S. curvatum*, a congener that has higher reproductive success. Therefore, the observed preference of *S. formosum* for Salticidae largely reduces the chances of them competing with *S. laetum*, another native *Sceliphron* species that prefers Araneidae. 

We observed that amongst Salticidae, *S. formosum* preferred certain genera, with 74% of Salticid prey belonging to the genera *Opisthoncus* and *Servaea*, which are both found on tree foliage in forests [31]. We therefore reason that *S. formosum* mainly preys on tree foliage or low vegetation surface in forests near the nesting sites, with forest environments playing a major role in facilitating predation for *S. formosum*. 

*S. formosum* also showed a preference for a specific prey size [5]. Despite being large spiders with adult body lengths reaching up to 70 mm, Sparassidae spiders found in mud nests were all juveniles and only had an average body length of 3.54 mm. In our study, we explain this difference by examining the body length to leg span ratio (BLR). Across the dominant spider families found in nests, salticid spiders have the largest body length to leg span ratio. Assuming leg span is a limiting factor in the capture or packing of the spider prey into the nest, spiders with a larger BLR will presumably provide more food mass for the developing larvae. Consequently, salticid spiders will represent the best choice from an efficiency perspective. 

Although foragers can be classified as generalists and specialists [32], it has become clear that a population of a generalist species may be made up of specialist individuals [33]. We have determined that *S. formosum* has a different prey preference from other *Sceliphron* species, but individual female wasps may also have specialized prey preferences [34]. We observed three mud nests that have specialized prey selection, one with 60% two-tailed spiders (Hersiliidae), one with 62% two-tailed spiders (Hersiliidae) and one with 75% huntsman spiders (Sparassidae). It is unlikely that these individuals are equipped with sensory organs to capture specific spider families, and so it may be explained by the nesting process, where females may tend to go after the first prey they find and keep hunting for similar prey or be determined by the local spider community.

Moreover, our data reveal that BLR is a useful new measure of spider morphology. Because *S. formosum* builds separated, single-cell mud nests, and usually has smaller nest cells than other *Sceliphron* species, *S. formosum* choses prey with a larger BLR to be able to store more prey in one nest. BLR is therefore a measure of space limitation in the wasp nest.

Lastly, our data might provide a preliminary study to further examine the interaction of urbanization and prey preference on spider fauna. Previous studies have shown that the *Sceliphron* species is a potentially important cause of spider mortality as most hunted spiders are juveniles and females, which can substantially reduce the reproductive success of targeted spiders [14]. On top of that, since *Sceliphron* mainly live in urban areas and utilize human constructions, with urbanization playing a big role in the ecological pattern [35], determining the spider composition [36], the prey preference of *Sceliphron* could be largely affected by it. We have documented a general prey preference in a largely unstudied species *S. formosum,* as well as intriguing individual specializations. These observations may form an important component in assessing the make-up and vulnerabilities of the spider fauna in urban environments. 

### 4.2. Nest Ecology

Prior to our study, parasitic insects of *Sceliphron* species had also been reported in multiple articles [37,38,39,40,41,42,43]. Our analyses have revealed that the nesting behaviour of a *Sceliphron* species provides a niche for many insect species in an urban ecosystem. Because *Sceliphron* species build nests on human constructions, people usually see them as pests or an annoyance, and nests are cleared away. However, mud nests may actually play an important role in urban ecosystems by hosting a diversity of insects.

A keystone species is one whose effect is disproportionally large relative to its abundance, by providing the major energy flow and three-dimensional structure that supports and shelters other organisms [44,45,46,47,48,49,50]. Thus, our study can serve as a preliminary study of whether *S. formosum* could be classified as a keystone species that provides a micro niche for urban–rural insect communities and maintains local biodiversity. Through the analysis of nest content, this study has produced 1. Novel host association records; 2. Notes on new species; 3. Records of Australian beneficial pollinators; and 4. Records of Australian native species. 

### 4.3. Host Associations

Arachnids, despite being generalist predators of arthropods, are known to have their own natural enemies, which include various groups of Hymenoptera and Diptera. In contrast to the great taxonomic diversity of hymenopteran spider enemies, fewer families of dipterans parasitise or prey upon arachnids [51]. Here, we discovered a new host association between *Ogcodes pygmaeus* (Diptera: Acroceridae) and *Servaea* Simon jumping spiders (Araneae: Salticidae) [31], which was made possible by a combination of predatory behaviour by *S. formosum* and parasitoid behaviour by *Ogcodes pygmaeus*. Acroceridae are cosmopolitan flies comprising approximately 550 species in 55 genera [52]. The largest genus, *Ogcodes* Latreille includes over 90 known species, with 25 found in Australia [53]. It is difficult to obtain their host records due to the rarity of witnessing them with their hosts in the wild, and museum specimens can only provide taxonomic information. In past studies, hosts of 21 species of *Ogcodes* have been recorded spanning 15 spider families [51] and 15 genera of Salticidae have been recorded as hosts among four acrocerid genera (*Ogcodes*, *Acrocera* Meigen, *Pterodontia* Gray and *Terphis* Erichson). *Ogcodes* is likely to be the major natural enemy of Salticidae [51]. Our discovery likely resulted from the overlapping prey preference of *Ogcodes pygmaeus* and *S. formosum*, as both species have salticid spiders as part of the larval diet. We consider these acrocerid flies to be “unlucky tenants” that initially parasitised their salticid host, which was subsequently captured by *S. formosum* and sealed inside the mud nests. This represents a new method for obtaining acrocerid fly host records and provides evidence of a novel host association for *O. pygmaeus* acrocerid flies and two species of *Servaea* Simon jumping spider hosts. 

Besides acrocerid flies, we also reported a host association of a new species of *Thraxan* beefly with its host *Pison simillimum,* and a detailed description and taxonomy has since been published by Li et al. [29]. This is an important discovery that reveals the larval development of beeflies. 

### 4.4. Notes on New Species 

The taxonomy of *Epipompilus* spider wasps (Hymenoptera: Pompilidae) is problematic. Not only are they rarely collected in the field, but they are also uncommon in museum collections [54]. The genus exhibits unusual and primitive structural traits within Pompilidae [22], and a primitive behaviour where females search larval hosts by crawling around spider habitats and laying eggs directly on prey, instead of constructing an underground nest to store paralysed spider prey. A new species of *Epipompilus* was found living inside the *Sceliphron* nest and, presumably, it is the primitive behaviour of *Epipompilus* that results in them inhabiting the same mud nest as *S. formosum.* There are two possible scenarios: the first is that a female *Epipompilus* found the wasp nest while it was still in the process of being provisioned by *S. formosum,* and, as an opportunistic female *Epipompilus,* laid an egg on one of the prey items inside the nest, hence making spider wasp and mud dauber wasp co-inhabitants. The second is the by-catch theory, which simply suggests that a *S. formosum* female captured a spider that had already been parasitised by *Epipompilus.* Because *Epipompilus* has been documented to be one of the only pompilid koinobionts (i.e., parasitoids that allow the host to continue developing) [55], it is likely that the by-catch scenario explains the presence of *Epipompilus* in the *Sceliphron* nest. 

We also provided information on an undescribed *Thraxan* species [29]; however, since we only successfully reared two females, a description is not possible because species identification is strongly based on male genitalia.

### 4.5. Australian Beneficial Pollinators 

Most of the tenants are flower visitors as adults, which means the mud nest potentially provides an important niche for pollinators. Adult *Sceliphron* feed on nectar and carry pollen on their bodies. A *Sceliphron* species has been suggested as the principal pollinator of endemic orchids in Madagascar [56] and sphecid wasps generally have been designated as one of the most efficient insect pollinators of carrots [57]. 

Besides sphecid wasps, two species of potter wasps (Eumeninae) were found to be reusing abandoned nests, representing the second largest group found from *Sceliphron* nests. Both species visit flowers as adults, and *Paralastor* eumenid wasps have been reported as pollinators for mango [58]. 

Moreover, amongst the pollinators found associated with mud nests, we discovered two species of bees that are listed as Australian beneficial pollinators by PaDIL (an initiative of the Australian Government of Agriculture, in collaboration with Museum Victoria, Plant Health Australia, the Department of Agriculture and Food Western Australia and the Plant Biosecurity Cooperative Research Centre), *Hylaeus nubilosus* (masked bee, Colletidae) and *Megachile aurifrons* (resin bee, Megachilidae). Megachilidae have been developed as crop pollinators [59] and determined as important pollinators for a socio-economically important tree species in the Sudanian region [60]; in crops such as alfalfa, no seeds will form in the absence of bees, and a species of *Megachile* has been developed extensively as a commercial pollinator [61]; species of *Hylaeus* (Colletidae) have been suggested as potential pollinators of carrots [62]. This evidence suggests that bees discovered in mud nests are some of the most well-known pollinators in agriculture and ecosystems that we highly rely on, so it is important that our survey is able to provide their records in an urban ecosystem. In addition, their association with the use of *Sceliphron* mud nests indicates that the survey of *Sceliphron* nest ecology is a reliable method to monitor populations of Australian beneficial pollinators. Therefore, this study has the potential to establish a new bio-indicator for pollinator biodiversity in urban–rural areas. 

### 4.6. Australian Native Species 

As an insect fauna survey, our study provides important records of Australian native species. Aside from native insects that are beneficial to humans, native insects of other species were also discovered in *Sceliphron* nests. 

*Pison* spp. were found to be the most abundant tenants that reuse the nests of *S. formosum*. *Pison* is a cosmopolitan genus that is best represented in Australia and South America with a third of the species (~160 species) occurring in Australia [23]. By comparing different nesting behaviours, cocoons and adult characteristics, we were able to identify six *Pison* wasp species: *P. simillimum, P. auriventre* and *P. spinolae,* which simply reoccupied the empty *S. formosum* nest; *P. priscum* that remodelled the nest into a compartmentalized nest with multiple cells; and *P. ruficorne* and *P. rufipes,* which built ball-like nests inside the original nest. Of these, only *P. auriventra* and *P. spinolae* produce cocoons with a crispy and hard texture. We are the first to establish the nesting behaviour, differences in cocoons and the association of *Pison* with *S. formosum.* Because the genus prefers to nest in pre-existing cavities [20], we suspect the high diversification within the genus has resulted from different nesting behaviours limited by environmental factors. Species distributed in different areas may have a particular preference in nesting sites, therefore leading to specific associations with certain abiotic factors (cavities in human structure, caves, etc.) or biotic factors such as *Sceliphron* nests. It would be interesting to see if *S. formosum* outside the study area host the same *Pison* species, and further determine if this association of nests can be influential in the evolution of *Pison*. 

We also documented gasteruptid wasps, satellite flies and eulophid wasps as native predators and parasitoids of Australian solitary bees and wasps. First, we provided the association of *Gasteruption cinerescens* (Gasteruptiidae) with the use of *S. formosum* nests. There are only nine genera of gasteruptid wasps worldwide and most species are found in Australia. To date, there are no recorded host records for this species. Nonetheless, through examining the nest contents we reared out, it is possible for us to further provide novel host association records between *G. cinerescens*, *Hylaeus nubilosus* and *Pison* sp. 

Second, we discovered the host association of *Amobia auriceps* and *S. formosum*, and reported the association of *Melittobia australica* (Eulophidae) and its hosts, which are associated with the use of mud nests, including *S. formosum* (Hymenoptera: Sphecidae)*, Pison* spp (Hymenoptera: Crabronidae) and *A. auriceps* (Diptera: Sarcophagidae). 

*Sceliphron* nests can also become bio-reservoirs of parasitic insects, and for which establishing host records can contribute to establishing new biological control agents. Three families of chalcid wasps, Eulophidae, Torymidae and Chalcididae were found in the nest. With most chalcid wasps being parasitoids, there are multiple successful biocontrol agents belonging to that group. For example, *Torymus sinensis* (Torymidae) is an effective control agent for invasive chestnut gall wasps [63,64]; a species of Toryminae (Torymidae) was also reported as a potential biocontrol agent against African fruit fly [65]; and *Brachymeria* species (Chalcididae) was reported as a potential biocontrol agent for serious banana pests [66]. Hence, our study has not only shed light on the local insect fauna, but also reported potential native biocontrol agents.

## 5. Conclusions

Recent studies of *Sceliphron* wasps have focused on their hunting behaviour, prey preference and nesting behaviour [13,14,67,68]. We have identified a novel and important new area of study involving *Sceliphron* wasp nest ecology and the impact it has on the local ecosystem by examining the tenants that use the nests of *Sceliphron.*


Our work shows how nest ecology may potentially serve as an important bio-indicator in which the *Sceliphron* wasp is a keystone species in an urban area, and an indicator of local biodiversity. Mud nests of *S. formosum* are excellent shelters for many insect species, and the nest itself is a micro niche for maintaining the local diversity of insect fauna. The loss of suitable habitat for nesting could affect the entire community of mud nests and possibly negatively influence pollination for many plant species. Hence, it is important to consider the role that *S. formosum* plays in urban areas. By studying the nest ecology, diverse tenants found in nests can provide important information in ecology, biodiversity research and environment monitoring. Because *Sceliphron* wasps are cosmopolitan, study methods of nest ecology similar to ours can contribute to biodiversity studies across the world.

## Figures and Tables

**Figure 1 insects-13-01136-f001:**
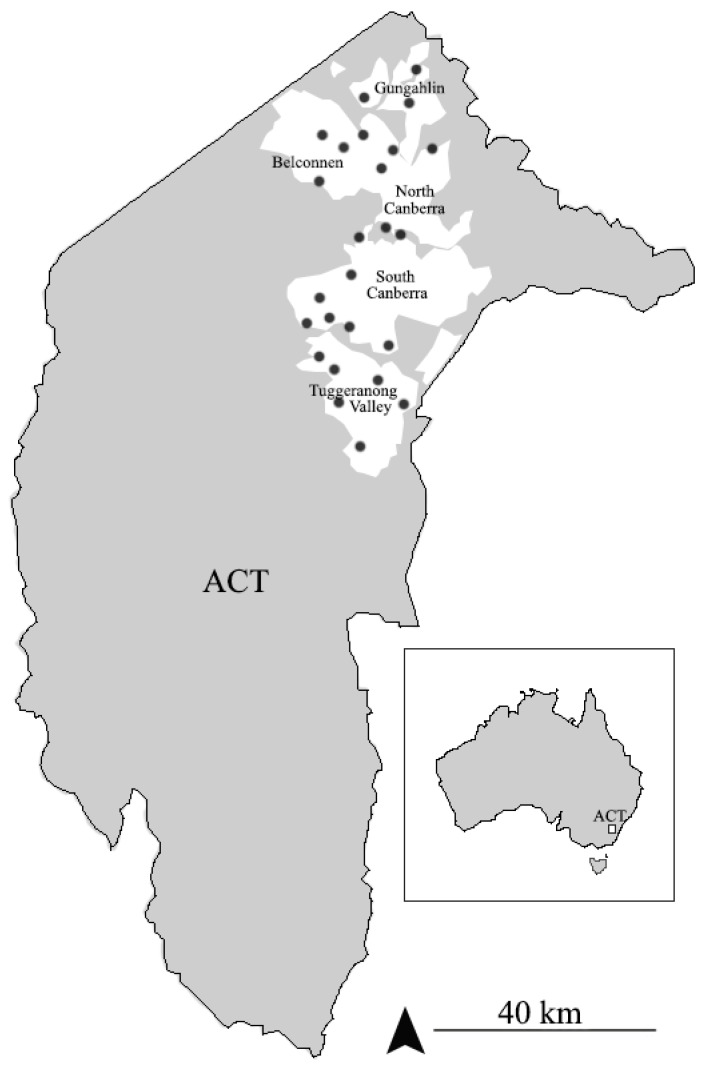
Nest collecting sites within ACT: urban area and Canberra suburbs highlighted with white and the nest collected sites are marked with black dots. (Total sites visited: 95.)

**Figure 2 insects-13-01136-f002:**
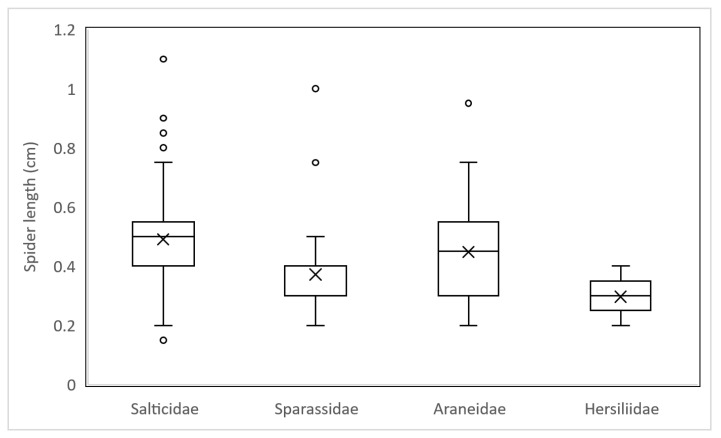
Spider body length for the four most abundant spider families found in mud nests of *Sceliphron formosum* (Hersiliidae, n = 29; Araneidae, n = 35, Sparassidae, n = 23; Salticidae, n = 515). Points of hollow represents outliers.

**Figure 3 insects-13-01136-f003:**
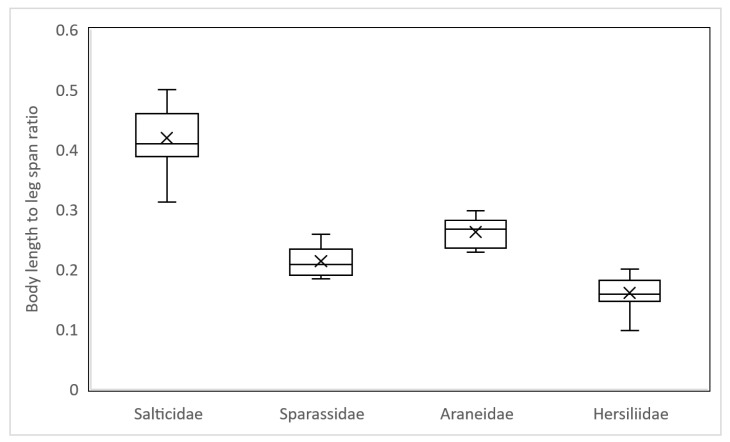
Body length to leg span ratio across four spider families found in mud nests of Sceliphron formosum (Hersiliidae, n = 20; Araneidae, n = 12, Sparassidae, n = 7; Salticidae, n = 17).

**Figure 4 insects-13-01136-f004:**
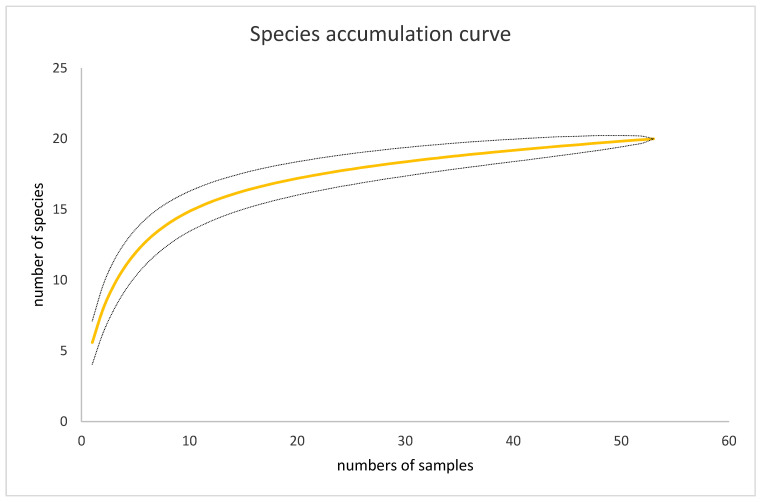
Species accumulation curve, represented with the yellow line and the 95% confidence interval dotted lines, displaying the data of 655 nests collected at 53 sites that contain 21 species of insect tenants that are associated with the use of mud nest.

**Figure 5 insects-13-01136-f005:**
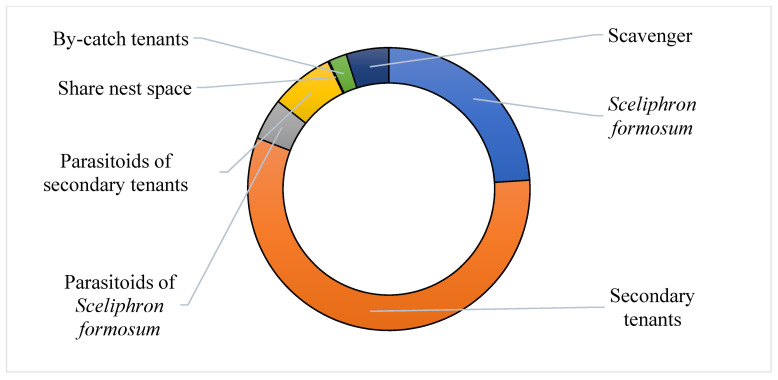
Percentage of *Sceliphron formosum* nests occupied by seven groups of insect tenants (n = 655): 1. Nest builder: *Sceliphron formosum* (24.7%) 2. Secondary tenants (58.6%) 3. Parasitoids of nest builders (4.8%) 4. Parasitoids of secondary tenants (7.4%) 5. Invader that shares the mud nest (0.1%) 6. By-catch tenants (2.1%) 7. Scavenger (5%).

**Figure 6 insects-13-01136-f006:**
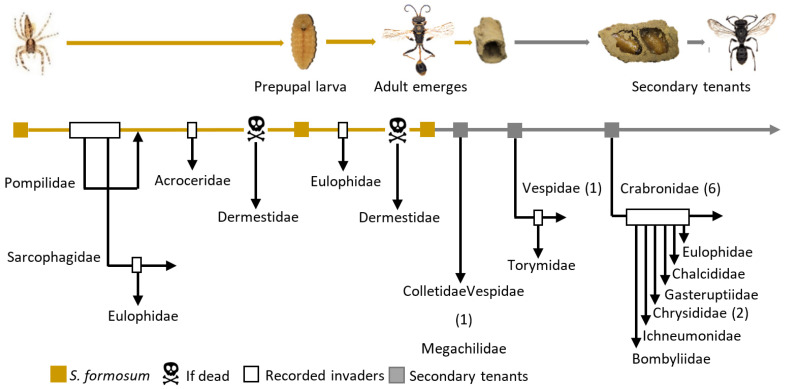
Insect tenants succession map within the life cycle of *Sceliphron formosum* mud nest. Yellow box represents the nest builder mud dauber, white box representing invasion event by parasitic insects, grey box representing the occupancy of new tenants following the emergence of mud dauber wasp adults, and skull represents the occupancy of new tenants following the death of mud dauber larvae. The number behind insect families indicates that more than one species was discovered within the family that is associated with the use of mud nests, and also shows the number of species involved at different time points.

**Figure 7 insects-13-01136-f007:**
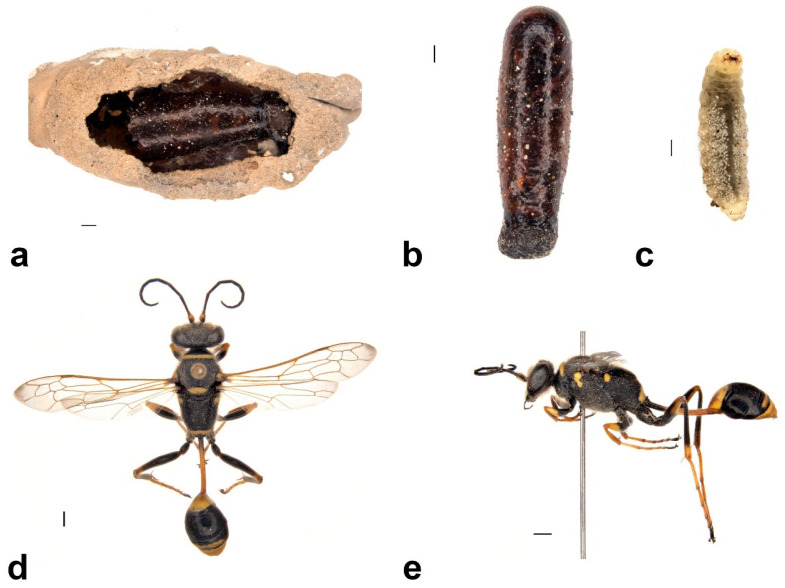
Larva, cocoon, mud nest and adult of *Sceliphron formosum*: (**a**) ventral view of the mud nest that contained emptied debris of cocoon, (**b**) cocoon that contained a nearly emerged pupa, (**c**) ventral view of a larva, (**d**) ventral view of male adult, and (**e**) lateral view of male adult. Scale bar = 1 mm.

**Figure 11 insects-13-01136-f011:**
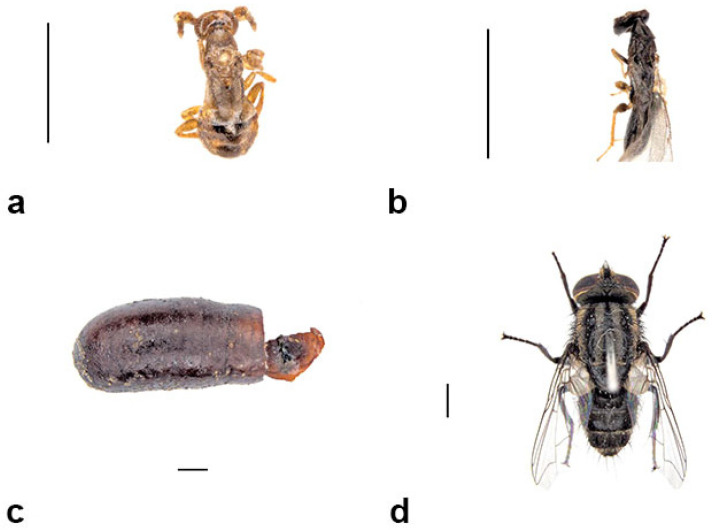
Main parasitoids of *Sceliphron formosum*, of which *Melittobia australica* was also seen parasitising other tenants: (**a**) dorsal view of male *M. australica* (**b**); dorsal view of female *M. australica;* (**c**) pupal case of *Amobia burnsi;* (**d**) dorsal view of *A. burnsi*. Scale bar = 1 mm.

**Figure 12 insects-13-01136-f012:**
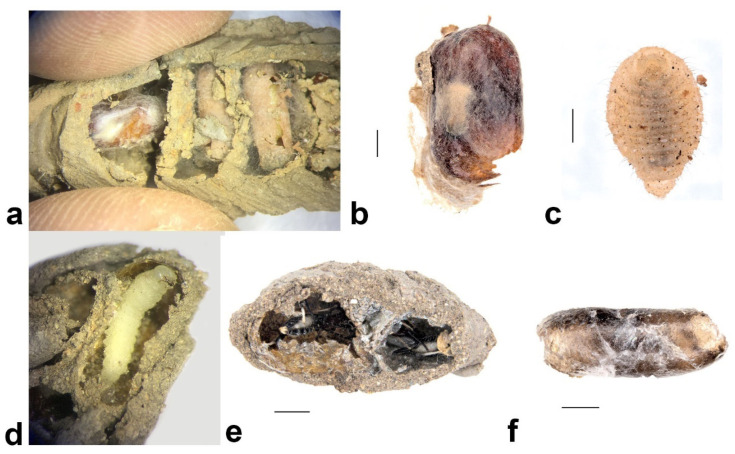
Evidence of the parasitoids of secondary tenants that were found in the *S. formosum* nest: (**a**) *Primeuchroeus reversus* in a nest of *Pison priscum:* cocoon of cuckoo wasp occupied the first cell on the left; (**b**) red bean-like cocoon of *Primeuchroeus faustus;* (**c**) ventral view of larval Toryminae wasp; (**d**) *Gasteruption cinerescens* larva in a nest of *Pison* sp.; (**e**) two nearly emerged *adult G. cinerescens* in a nest of *Hylaeus nubilosus*, with visible cellophane coatings, assuming their host is *Hylaeus nubilosus;* (**f**) brown cocoon *of Phrudus* sp. Scale bar = 1 mm.

**Figure 13 insects-13-01136-f013:**
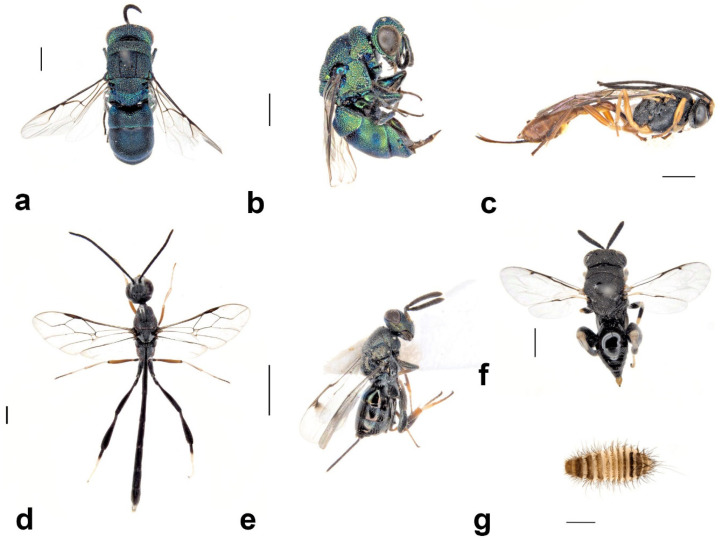
Group 4, Adult parasitoids of the secondary tenants (**a**–**e**), and Group 7, the Scavenger (**f**): (**a**) dorsal view of *P. reversus;* (**b**) lateral view of *P. faustus*; (**c**) lateral view of *Phrudus* sp.; (**d**) dorsal view of male *G. cinerescens;* (**e**) lateral view of Toryminae wasp; (**f**) dorsal view of *Brachymeria* sp.; (**g**) dorsal view of *Anthrenus* larva. Scale bar = 1 mm.

**Table 1 insects-13-01136-t001:** Spider composition of 43 nests, including at least seven spider families.

Spider Family	Numbers of Individuals in Nests
Salticidae	514 (82.5%)
Araneidae	42 (6.7%)
Hersiliidae	29 (4.7%)
Sparassidae	23 (3.7%)
Thomisidae	8 (1.3%)
Amaurobiidae	1 (0.2%)
unknown	5 (0.8%)

**Table 2 insects-13-01136-t002:** Identification of 265 salticid spiders collected from mud nests of *Sceliphron formosum*, including at least 11 genera and 15 species of Salticidae.

Spider Prey	Number of Spiders
*Opisthoncus parcidentatus*	94 (35.5%)
*Opisthuncus* sp.	21 (7.9%)
*Servaea narraweena*	59 (22.3%)
*Servaea villosa*	16 (6%)
*Servaea incana*	5 (1.9%)
*Servaea* sp.	1 (0.4%)
*Cytaea* sp.	38 (14.3%)
*Helpis* sp.	8 (3%)
*Simaethula* sp.	7 (2.6%)
*Holoplatys* sp.	4 (1.5%)
*Simaetha* sp.	3 (1.1%)
*Zenodorus* sp.	1 (0.4%)
*Bianor maculatus*	1 (0.4%)
*Sandalodes* sp.	1 (0.4%)
*Clynotis* sp.	1 (0.4%)
Other	5 (1.9%)
Total	265

**Table 3 insects-13-01136-t003:** Seven groups of insect tenants are associated with the use of mud nests.

Type of Tenant	Family or Species	Number of Nests Occupied
**1. Nest builder**	*Sceliphron formosum* (Sphecidae)	162
**2. Secondary tenants**	*Pison* spp. (Crabronidae)	266
	Eumeninae (Vespidae)	101
	*Hylaeus nubilosus* (Colletidae)	15
	*Megachile aurifrons* (Megachilidae)	2
**3. Parasitoids of *S. formosum***	*Amobia burnsi* (Sarcophagidae)	12
	*Melittobia australica* (Eulophidae)	20 (type 3 + 4)
**4. Parasitoids of secondary tenants**	*Melittobia australica* (Eulophidae)	20 (type 3 + 4)
	Toryminae (Torymidae)	1
	*Brachymeria* sp. (Chalcididae)	1
	*Phrudus* sp. (Ichneumonidae)	1
	*Gasteruption cinerescens* (Gasteruptidae)	7
	*Primeuchroeus faustus* (Chrysididae)	5
	*Primeuchroeus reversus* (Chrysididae)	2
	*Thraxan* sp. (Bombyliidae)	5
**5. Share space with *S. formosum***	*Epipompilus* sp. (Pompilidae)	1
**6. By-catch**	*Ogcodes pygmaeus* (Acroceridae)	14
**7. Scavenger**	*Anthrenus* sp. (Dermestidae)	33

## Data Availability

The data presented to support the findings of this study are contained within the article.

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
