# Peer review of "Nest Ecology and Prey Preference of the Mud Dauber Wasp Sceliphron formosum (Hymenoptera: Sphecidae)"

_insects, 2022, doi:10.3390/insects13121136_

Round 1
Reviewer 1 Report
This is an excellent, carefully done study of the nest ecology of an Australian species of the mud-dauber genus Sceliphron. Mud-nests by Sceliphron are are large and conspicuous and have been repeatedly collected and reared in the lab. In this respect, the general methodology of the current study is quite similar, but with the important difference, that the sample size is impressive. I don't know of any other study of Sceliphron nests, where so many nests have been collected anmd analized. The large sample size is a prerequisite to study the complex relationships between the nest builder and alls the parasitoids and to set up sound statistical analyses. So I am generally very impressed by the manuscript and can only suggest to publish it as soon as possible.
My only minor concern is that the papers misses some published information from other species in Sceliphron. My quick search of the Sceliphron file of Pulawski's Catalog of Sphecidae reveals a few publications, which I think should be added for comparative purposes. An example is Pham, Truong, and Nguyen, 2019:42 (on a nest parasite of Sceliphron madraspatanum in Vietnam). By the way, reference #42, the family name is Pham, not Huy. So I strongly suggest to carefully check Pulawski's references for Sceliphron for other papers on the nesting ecology in Sceliphron (https://researcharchive.calacademy.org/research/entomology/entomology_resources/hymenoptera/sphecidae/genera/Sceliphron.pdf).
Otherwise an excellent manuscript!
Author Response
Responses to reviewers’ feedback
Thank you for your time and all the feedbacks!! Please see our revisions from below:
- So I strongly suggest to carefully check Pulawski's references for Sceliphron for other papers on the nesting ecology in Sceliphron:
I have found another similar paper about Sceliphron caementarium nest ecology and I have included it in the introduction:
On the other hand, Sceliphron nest ecology has not been noticed, yet comprehensively studied. Only few studies and observations in the past recorded that mud nests, despite of its breeding purposes, are often exploited by parasitic insects or opportunistic insects that use mud nests for reproduction, and most of which are about the Sceliphron species under subgenus Sceliphron, which build multi-cell mud nests. For example, nests of S. laetum have been reported as being parasitised by flies of the families of Bombyliidae, Sarcophagidae, and wasps of the families Eulophidae and Chrysididae; while abandoned nests can be used by wasps of the families Crabronidae and Vespidae [47]. Nest of S. caementarium were reported in Italy with their parasitoids, inquilines and parasitoids of the inquilines [18]. In Crimea, a recent study reported that nests of Sceliphron destillatorium constitute an important resource for insect species that nest in pre-existing cavities [30].

Reviewer 2 Report
A very interesting paper! I myself have witnessed mud dauber wasp nest parasitism in California, USA. I’ve always wondered to what extend these nests support biodiversity. Generally, I think the paper is strong. I do think the weakest part of the paper exists in the introduction and with the landscape analysis.
Firstly, there are a lot of awkward transitions in the introduction. Lines 37, 46, and 48 are good examples. I think some slight re-writing here could resolve that.
Line 64: What are cues? Are you using them as an explanatory variable? The term seems informal and this concept seems like it is not connected throughout the rest of the paper.
Line 76: “ACT” is not introduced (I am sure it makes sense regionally!)
Up to line 79: There is a lot of information in the introduction that seems unnecessary? If you are just looking at prey preference and ecology, why do we need to know about web building, etc.? I think this could be connected more to the rest of the paper throughout.
81: What is a breeding cradle? I am not familiar with this term.
Line 80: Maybe bring this forward? Same with lines 159-163. If the focus is the nests, the spiders, and associated organisms, tell us more about that in the intro.
142-143: I think you need to introduce this in the introduction. What resolution is this data? I think it would be hard to argue that vegetation had a realistic effect if the resolution is too wide. Also, this range seems unrealistic. How could a forest 1 km away affect prey selection? How far do these wasps forage? I think you do cover some of this in lines 582-596, but it still seems like a bit of a stretch to assign non-forested land as “urban” when we know a wide variety of land types and management practices exist within these areas. Secondly, I think if you did keep this in the paper it would require some significant additions. What affects does urbanization have to spiders and wasps (or species richness/evenness)? Why? What extend of urbanization induces these effects?
159-169: This seems like it should move to the introduction unless you completed some analysis on this data, in which case only results of analysis or survey work should be reported here.
Figure 2: It’s hard to understand the median of this data w/o some bolding of the median line.
3.3.1: Generally, as above, I am skeptical of the importance of these results. Too many confounding factors exist here, especially in the context of an urban area. We know species composition and distribution in urban areas can be greatly impacted with very small-scale changes. A lot of information exists here in the context of urban agriculture. See Dr. Monika Egerer and Dr. Stacy Philpott’s work out of UC Santa Cruz. So many factors in play here. In my opinion, this aspect of the paper should be omitted, unless ground-proofing was accomplished, or some other characteristics of sampling sites were included. Also, I think it would make the paper stronger. A stronger focus on the nest and wasp ecology is valuable and compelling.
Figure 8 is rendering very poorly…
Figure 9: Scale bar on each picture is distracting. Not sure if there is a fix here.
I would strike lines 595-597. Too many confounding factors here.
623-624: Does it though? Are these generalist or specialists? Are these nests required for their persistence in an urban area? I think 626-627 is a valid and important point.
628-632: I think this is a very bold claim and I am not sure your data supports this conclusion.
686: Carrots or Daucus spp.?
Conclusion: As previously stated, a very strong and interesting paper. I think the intro needs a bit more work to help the rest of the paper flow. I would get rid of the landscape analysis - too general and does not add much to the paper. I think that would be an interesting follow up paper/research.
Thanks for the opportunity to read!
Author Response
Responses to reviewers’ feedback
Thank you for your time and all the feedbacks!! Please see our revisions from below:
- Firstly, there are a lot of awkward transitions in the introduction. Lines 37, 46, and 48 are good examples. I think some slight re-writing here could resolve that.
re-written
- Line 64: What are cues? Are you using them as an explanatory variable? The term seems informal and this concept seems like it is not connected throughout the rest of the paper.
re-written
- Line 76: “ACT” is not introduced (I am sure it makes sense regionally!)
Full name of ACT, Australian Capital Territory added!
- Up to line 79: There is a lot of information in the introduction that seems unnecessary? If you are just looking at prey preference and ecology, why do we need to know about web building, etc.? I think this could be connected more to the rest of the paper throughout.
deleted unnecessary information
- Line 81: What is a breeding cradle? I am not familiar with this term.
re-written, using breeding purposes instead
- Line 80: Maybe bring this forward? Same with lines 159-163. If the focus is the nests, the spiders, and associated organisms, tell us more about that in the intro.
rearranged and rewritten the paragraphs
- Line 142-143: I think you need to introduce this in the introduction. What resolution is this data? I think it would be hard to argue that vegetation had a realistic effect if the resolution is too wide. Also, this range seems unrealistic. How could a forest 1 km away affect prey selection? How far do these wasps forage? I think you do cover some of this in lines 582-596, but it still seems like a bit of a stretch to assign non-forested land as “urban” when we know a wide variety of land types and management practices exist within these areas. Secondly, I think if you did keep this in the paper it would require some significant additions. What affects does urbanization have to spiders and wasps (or species richness/evenness)? Why? What extend of urbanization induces these effects?
deleted
- Line 159-169: This seems like it should move to the introduction unless you completed some analysis on this data, in which case only results of analysis or survey work should be reported here.
Only observations from fieldworks were kept, title added
- Figure 2: It’s hard to understand the median of this data w/o some bolding of the median line.
mean markers in the middle
- 3.1: Generally, as above, I am skeptical of the importance of these results. Too many confounding factors exist here, especially in the context of an urban area. We know species composition and distribution in urban areas can be greatly impacted with very small-scale changes. A lot of information exists here in the context of urban agriculture. See Dr. Monika Egerer and Dr. Stacy Philpott’s work out of UC Santa Cruz. So many factors in play here. In my opinion, this aspect of the paper should be omitted, unless ground-proofing was accomplished, or some other characteristics of sampling sites were included. Also, I think it would make the paper stronger. A stronger focus on the nest and wasp ecology is valuable and compelling.
deleted
- Figure 8 is rendering very poorly…
re-rendered
- Figure 9: Scale bar on each picture is distracting. Not sure if there is a fix here.
- I would strike lines 595-597. Too many confounding factors here.
deleted
- Line 623-624: Does it though? Are these generalist or specialists? Are these nests required for their persistence in an urban area? I think 626-627 is a valid and important point.
Generalists so we wouldn’t be able to tell it’s persistence regarding mud nests. Re-written
- Line 628-632: I think this is a very bold claim and I am not sure your data supports this conclusion.
Re-written
- Line 686: Carrots or Daucus spp.?
Only mentioned carrots in the article
Re-written introduction:
Mud dauber wasps belong to the sphecid genus Sceliphron Klug, which contains 35 species occupying all major biogeographical regions of the world [42]. Three species have been recorded in Australia: the two endemic species Sceliphron laetum Smith and Sceliphron formosum Smith, and the introduced Sceliphron caementarium Drury, from North America [46]. In natural situations, Sceliphron nests are constructed on shaded and sheltered substrates, such as rock overhangs, sheltered sites on trees or in hollow logs [46]. While in urban areas, Sceliphron wasps are seen commonly inhabiting human constructions, building mud nests under eaves and roofs, along the periphery of windows and in other sites that provide the necessary shelter [8]. The mud nest is the brooding room for their larvae, consisting of one or multiple cells that are provisioned with paralysed spiders [4]. The majority of mud dauber wasps practice mass provisioning, where female wasps prepare food for larvae prior to laying their eggs [25].
While all Sceliphron species practice similar larval provisioning, their nests may differ in shape and in number of mud cells. Wasps in the subgenus Sceliphron construct multiple mud cells that amalgamate into one large mud nest, whereas those in the subgenus Prosceliphron, build single-celled mud nest [40]. Of the three Sceliphron species that occur in Australia (Sceliphron formosum, Sceliphron laetum and Sceliphron caementarium), S. formosum is the only member of the subgenus Prosceliphron and is the least studied. While S. laetum is widespread in Australia, S. formosum is more confined to northern and eastern Australia and less spotted elsewhere [16]. Despite their rarity elsewhere, mud nests of S. formosum are common in urban areas of the Australian Capital Territory (ACT).
To date, most studies regarding Sceliphron wasps are of its prey preference that associates to the larval provisioning and the wasp-spider interaction. Despite of an early study proposing that Sceliphron species hunt spiders without any prey preference [2], there is now enough evidence to argue that there is prey preference shown by Sceliphron, and it is affected by spider sizes [25], spider taxa [3, 13, 17, 25, 29, 51, 52, 64, 66], spider defense responses [12] and individual specialization [53]. Among which the prey preference of spider taxa was the most frequently discussed. Though studies have consistently shown that Sceliphron species prefer to prey on orb-web spiders (Araneidae) [3, 13, 17, 25, 29, 51, 52, 64, 66], a single observation [16] recorded Sceliphron formosum having preferred prey largely composed of jumping spiders (Salticidae). To further explore this observation, the first part of our study aimed to record the prey preference of S. formosum in ACT, as well as investigating the drivers behind its specific prey preference.
On the other hand, Sceliphron nest ecology has not been noticed, yet comprehensively studied. Only few studies and observations in the past recorded that mud nests, despite of its breeding purposes, are often exploited by parasitic insects or opportunistic insects that use mud nests for reproduction, and most of which are about the Sceliphron species under subgenus Sceliphron, which build multi-cell mud nests. For example, nests of S. laetum have been reported as being parasitised by flies of the families of Bombyliidae, Sarcophagidae, and wasps of the families Eulophidae and Chrysididae; while abandoned nests can be used by wasps of the families Crabronidae and Vespidae [46]. Nest of S. caementarium were reported in Italy with their parasitoids, inquilines and parasitoids of the inquilines[]. In Crimea, a recent study reported that nests of Sceliphron destillatorium constitute an important resource for insect species that nest in pre-existing cavities [29].
Of great interest are data on the influence of mud nests of S. formosum on the native insect fauna. Here we refer to those insects that use cells of S. formosum as “tenants” due to the nature of them using the pre-existing mud nests. Second part of our study presents the discovery through the analysis of more than 650 mud nests of S. formosum and their contents, uncovering the nest ecology, the community dynamics and tenant succession associated with the S. formosum mud nest system.
Re-written conclusion:
Prior to our study, parasitic insects of Sceliphron species had also been reported in multiple articles [6, 31, 32, 33, 41, 37, 62].Our analyses have revealed that the nesting behaviour of a Sceliphron species provides a niche for many insect species in an urban ecosystem. Because Sceliphron species build nests on human constructions, people usually see them as pests or an annoyance, and nests are cleared away. However, mud nests may actually play an important role in urban ecosystems by hosting a diversity of insects.
A keystone species is one whose effect is disproportionally large relative to its abundance, by providing the major energy flow and three-dimensional structure that supports and shelters other organisms [1, 22, 24, 34, 35, 49, 61]. Thus, our study can serve as a preliminary study of whetherS. formosum could be classified as a keystone species which provides a micro niche for urban-rural insect communities and maintains local biodiversity. Through the analysis of nest content, this study has produced: 1. Novel host association records; 2. Notes on new species; 3. Records of Australian beneficial pollinators; and 4. Records of Australian native species.

Round 2
Reviewer 2 Report
A really nice improvement. Overall I think it is ready to go. I had a few observations:
Line 9: Italicize genus species
Figure 1 rendering poorly
Table 1 and 2: Nit-picky, but it would look better if they lined up (or perhaps were incorporated into one table?
600 strike “mysterious”
Author Response
Responses to reviewers’ feedback
Thanks again for your time and all the feedbacks!! Please see our revisions from below:
- Line 9: Italicize genus species
italicized
- Figure 1 rendering poorly
Modified
- Table 1 and 2: Nit-picky, but it would look better if they lined up (or perhaps were incorporated into one table?
lined up
- 600 strike “mysterious”
striked
